Methods

# Ribosomal stalling landscapes revealed by high-throughput inverse toeprinting of mRNA libraries

Britta Seip, Guénaël Sacheau, Denis Dupuy, C Axel Innis

**Although it is known that the amino acid sequence of a nascent polypeptide can impact its rate of translation, dedicated tools to systematically investigate this process are lacking. Here, we present high-throughput inverse toeprinting, a method to identify peptide-encoding transcripts that induce ribosomal stalling in vitro. Unlike ribosome profiling, inverse toeprinting protects the entire coding region upstream of a stalled ribosome, making it possible to work with random or focused transcript libraries that efficiently sample the sequence space. We used inverse toeprinting to characterize the stalling landscapes of free and drug-bound *Escherichia coli* ribosomes, obtaining a comprehensive list of arrest motifs that were validated in vivo, along with a quantitative measure of their pause strength. Thanks to the modest sequencing depth and small amounts of material required, inverse toeprinting provides a highly scalable and versatile tool to study sequence-dependent translational processes.**

## Introduction

Multiple factors determine the rate at which a nascent polypeptide is synthesized by the ribosome, among which is its amino acid sequence (Ito, 2014). Striking examples of translation regulation by the nascent chain are arrest peptides, a diverse class of regulators sharing a common propensity to block the ribosomes responsible for producing them (Wilson & Beckmann, 2011; Ito & Chiba, 2013; Wilson et al, 2016). In some cases, ribosome inhibition requires increased levels of a specific metabolite or drug, with the arrest peptide and ribosome acting as a combined sensor for these small molecules (Ramu et al, 2009; Vázquez-Laslop et al, 2011). More generally, simple features such as stretches of positively charged residues or proline-rich motifs can induce pauses in translation (Lu & Deutsch, 2008; Peil et al, 2013; Woolstenhulme et al, 2013; Qi et al, 2018). Ribosome-targeting antibiotics, such as macrolides, phenicols, or oxazolidinones, also slow down the translation of short problematic amino

acid motifs (Arenz et al, 2014; Davis et al, 2014; Kannan et al, 2014; Marks et al, 2016; Ramu et al, 2009; Vázquez-Laslop et al, 2010).

Ribosomal stalling on the mRNA induced by nascent peptides can impact processes such as co-translational protein folding or secretion (Nakatogawa & Ito, 2001; Chiba et al, 2009), mRNA degradation (Chiba et al, 1999; Chiba et al, 2003), or the expression of neighboring ORFs on the same transcript (Ito & Chiba, 2013; Wilson et al, 2016). Although the true impact of the nascent amino acid sequence on gene expression remains unknown, the large number of translated upstream ORFs discovered in recent years suggests that peptide-induced ribosomal stalling could represent a widespread form of translational control (Andrews & Rothnagel, 2014; Storz et al, 2014; Seip & Innis, 2016; Ndah et al, 2017). Moreover, understanding the link between the sequence-dependence of antibiotics and their efficacy as ribosomal inhibitors could help pinpoint the most promising lead compounds for further development. This is particularly true for macrolide antibiotics, which have seen a resurgence following the introduction of a total synthesis approach allowing the production of a virtually endless number of derivatives (Seiple et al, 2016; Dinos, 2017). Thus, efforts are needed to systematically investigate the complex interplay between nascent amino acid sequences and the rate at which they are synthesized under different environments.

At present, ribosome profiling (Ingolia et al, 2009) is the main tool capable of providing a global in vivo view of ribosome density on the mRNA, which can be used as a proxy for measuring sequence-dependent translation inhibition. Ribosome profiling has been used to study the translational pausing landscape in bacteria lacking the rescue factor elongation factor P (EF-P) (Woolstenhulme et al, 2015) and to identify nascent amino acid motifs responsible for antibiotic-dependent translational arrest in the Gram-negative bacterium *Escherichia coli* (Kannan et al, 2014) and in the Gram-positive bacterium *Staphylococcus aureus* (Davis et al, 2014). Ribosome-protected mRNA footprints obtained by ribosome profiling are short and encode only a few amino acids upstream of the pause site (Woolstenhulme et al, 2015; Mohammad et al, 2016). This means that information concerning the nascent peptide's amino acid sequence must be inferred from the mapping of ribosome footprints to a reference genome. Consequently, ribosome profiling cannot be used on a pool of uncharacterized coding sequences, as might be found in

Institut Européen de Chimie et Biologie, Université de Bordeaux, Institut National de la Santé et de la Recherche Médicale and Centre National de la Recherche Scientifique, Pessac, France

Correspondence: axel.innis@inserm.fr; denis.dupuy@inserm.fr

a hypothetical Systematic Evolution of Ligands by Exponential En-richment (SELEX)-type experiment (Ellington & Szostak, 1990; Tuerk & Gold, 1990) seeking to identify a novel metabolite-dependent arrest peptide. A couple of genetic selection techniques in bacteria have sought to overcome this drawback, but they are limited in scope (Tanner et al, 2009; Woolstenhulme et al, 2013). In addition, the bac-terial culture volumes and sequencing depth necessary to perform ribosome profiling are incompatible with the systematic screening of potential antibiotic leads or small molecules likely to act as co-inducers of arrest. To overcome these limitations, we developed a new, highly scalable in vitro method to investigate ribosomal stalling by nascent peptides encoded within transcript libraries of any given complexity.

Inverse toeprinting is a versatile selection strategy that relies on a highly processive 3′ to 5′ RNA exonuclease to degrade the mRNA downstream of the leading ribosome on a transcript. This makes it possible to determine the position of stalled ribosomes on the mRNA with codon resolution, while protecting the entire upstream peptide-encoding region. Inverse toeprinting is amenable to high-throughput, as next-generation sequencing can be used as readout. Because it is an in vitro method, it allows the precise control of translation conditions and can further be incorporated into se-lection schemes tailored to a variety of applications. We used in-verse toeprinting to explore the stalling landscapes of free and drug-bound bacterial ribosomes engaged in the translation of random and focused transcript libraries, and demonstrate its usefulness for rapidly assaying context-specific translation in-hibition by ribosome-targeting compounds.

# Results

### Inverse toeprinting maps stalled ribosomes with codon resolution while preserving the upstream coding region

Ribosome profiling generates short footprints and thus loses se-quence information for most of the coding region, whereas classical toeprinting (Hartz et al, 1989; Orelle et al, 2013) determines the exact position of a ribosome on the mRNA, but is not suitable for large-scale analyses (Fig 1A). By contrast, inverse toeprinting relies on RNase R, a 3′ to 5′ RNA exonuclease that efficiently degrades mRNA with a 3′ poly–(A) tail (Vincent & Deutscher, 2006). Our assumption was that RNase R would digest the mRNA up to a precise, discrete position downstream of the P-site codon of the foremost stalled ribosome-nascent chain complex. Inverse toeprints obtained in this manner could be analyzed by deep sequencing or evolved through multiple rounds of selection (Figs 1B and S1A).

As a proof-of-principle for inverse toeprinting, we generated a 5′–biotinylated (Fig S1B) and 3′–polyadenylated (Fig S1C) mRNA template encoding the erythromycin resistant methyltransferase B leader peptide (ErmBL), followed by a short coding region ending with a UGA stop codon. This template was used to express ErmBL in a PURE E. coli translation system (Shimizu et al, 2001) in the absence of release factor 2 (RF-2), which promotes peptide release at UGA or UAA stop codons (Scolnick et al, 1968), and in the absence or presence of the macrolide antibiotic erythromycin (Ery). ErmBL

undergoes translational arrest in an antibiotic-dependent manner (Arenz et al, 2014; Vázquez-Laslop et al, 2010), a property that is used in vivo to induce the expression of a methyltransferase gene on the same mRNA that confers resistance to macrolide antibiotics (Horinouchi & Weisblum, 1980). We chose ErmBL because it has been extensively characterized biochemically (Min et al, 2008; Gupta et al, 2016) and its structure has been determined within the context of a drug-stalled ribosome (Arenz et al, 2014; Arenz et al, 2016). In the presence of Ery, ribosomes undergo strong translational arrest when codon 10 of ermBL is in the ribosomal P-site, whereas in the absence of the drug ribosomes reach the UGA stop codon and stall because of the absence of RF-2. RNase R treatment yielded either short or long 3′–truncated mRNA fragments for ribosomes arrested at these two positions (Figs 1C and S1C). Ribosome-protected mRNAs were ligated to a 3′ linker enabling reverse transcriptase priming. Ribosomes stalled at the UGA stop codon protected an EcoRV site on the mRNA, which could be cleaved after reverse transcription and second strand synthesis. Thus, only short cDNA fragments derived from messengers encoding sequences that caused drug-dependent translational ar-rest were amplified by PCR after EcoRV treatment (Fig 1C). Similar experiments performed with other arrest peptides (ErmAL1, ErmCL, ErmDL, SecM, and TnaC) showed that inverse toeprinting can be used as a general tool for selecting arrest sequences (Fig 1D). Moreover, we showed that it is possible to perform consecutive rounds of selection by alternating restriction enzymes on the 3′ oligonucleotide linkers (Fig S1D). This could be used in the future as the basis for a SELEX-like (Ellington & Szostak, 1990; Tuerk & Gold, 1990) scheme to identify rare arrest sequences contained within a complex transcript pool.

To test the scalability of our method for high-throughput assays, we performed inverse toeprinting on an in vitro translation reaction using a template library ($NNS_{15}$ library) encoding 20-residue pep-tides with a variable region of 15 NNS codons, where N and S denote equal proportions of the four possible nucleotides or of G or C, respectively (Fig S2A). We size-selected inverse toeprints to mini-mize contamination from DNA fragments resulting from initiation complexes (Fig S3). The accumulation of initiation complexes was probably due to initiation being the rate-limiting step of translation and is also observed in the form of increased ribosomal density corresponding to initiation complexes in ribosome profiling ex-periments (Woolstenhulme et al, 2015). Removing these fragments ensured that more useful reads could be sequenced and could not introduce significant bias into our analysis. Paired-end Illumina sequencing revealed a tri-nucleotide periodicity for fragments where RNase R cleavage had occurred 24–47 nucleotides down-stream of the start codon (Fig 1E). Longer fragments did not follow this size periodicity and were excluded from our analysis.

UAG stop codons were enriched 1.4-fold after inverse toeprinting, indicating that ribosomes pause on these codons, despite the presence of release factor 1 (RF-1) (Scolnick et al, 1968) in the translation reaction. This pausing was used to precisely characterize the protective effect of stalled ribosomes on the mRNA. Given that ribosomes pause when they encounter a UAG stop in the A-site, our analysis revealed a distinct three-nucleotide peak positioned +17 nucleotides downstream from the ribosomal P-site (Fig 1F). This RNase R cleavage pattern was also confirmed for all of the arrest peptides we tested, allowing us to locate a second stalling site for the drug-dependent arrest peptide ErmAL1 that had been overlooked in

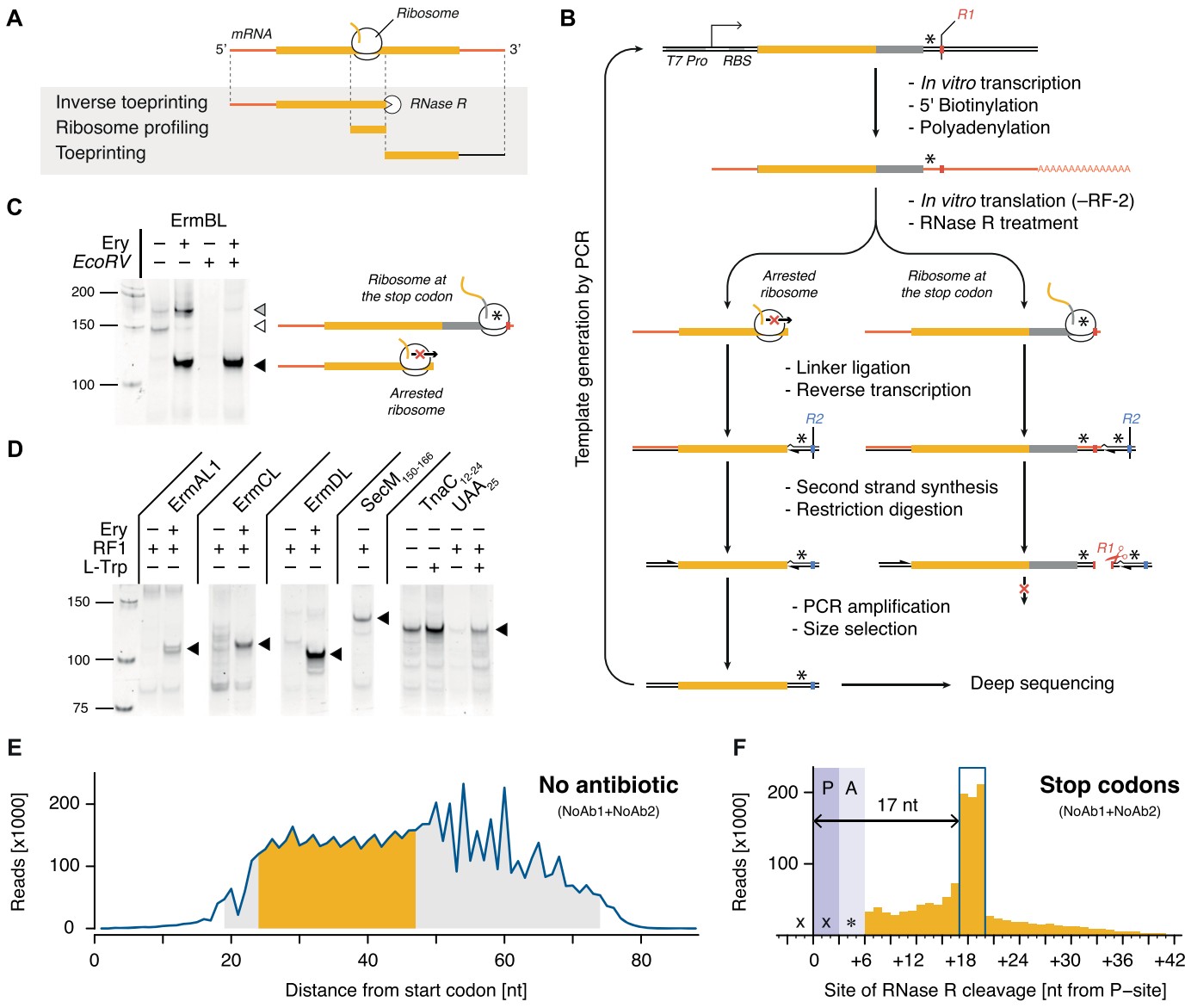

**Figure 1. Inverse toeprinting locates ribosomes on the mRNA with codon resolution.**
**(A)** Comparison between inverse toeprinting, ribosome profiling, and classical toeprinting. **(B)** Schematic overview of the inverse toeprinting workflow. Restriction enzymes used in odd (*EcoRV*) and even (*ApoI*) cycles are shown in red and blue, respectively. Stop codons are indicated as asterisks. **(C)** Removal of inverse toeprints featuring ribosomes that have reached the stop codon on the *ermBL* template (white triangle) using the *EcoRV* restriction enzyme. The black triangle corresponds to arrested ribosomes and the gray triangle to full-length mRNA. **(D)** Inverse toeprints for various Erm peptides in the absence or presence of Ery, SecM$_{150-166}$, and TnaC$_{12-24}$UAA$_{25}$. The wild-type UGA$_{25}$ stop codon for TnaC was replaced with a UAA$_{25}$ stop codon, allowing its release by RF-1. **(E)** Size distribution of inverse toeprints from two biological replicates (NoAb1 and NoAb2) with a minimum Q-score of 30 obtained from an NNS$_{15}$ library translated in the absence of any added antibiotic. The fragment size range shaded in gray corresponds to the band that was cut from a 12% TBE-acrylamide gel, whereas the range in yellow indicates fragments that were used in the subsequent analysis. **(F)** Analysis of inverse toeprints containing stop codons that were obtained in the absence of antibiotic reveals that RNase R cleaves +17 nucleotides downstream from the P-site.

previous studies (Ramu et al, 2011; Vázquez-Laslop et al, 2010) (Fig S4A–F). Thus, inverse toeprinting can be used to determine the position of stalled ribosomes on the mRNA at codon resolution.

## Motif enrichment correlates with pause strength and in vivo data

To assess whether sequences enriched following inverse toeprinting correlated with their ability to induce translational arrest or pausing, we first estimated the reproducibility of the frequency

of occurrence of 3-aa motifs of the type $X_1X_2(X_3)$ (where $X_2$ is in the P-site and $X_3$ in the A-site) between independent biological replicates. From ~3.4 million inverse toeprints obtained after translation in the absence of antibiotics, we could precisely (Fig S5) and reproducibly measure the frequency of 5,278 of 8,000 possible 3-aa motifs ($R^2$ = 0.95; <15% error between biological replicates) (Fig 2A). To limit the impact of the noise resulting from poor counting statistics, we chose to limit our subsequent analysis to this subset of well-measured 3-aa motifs. Because the vast majority of 3-aa

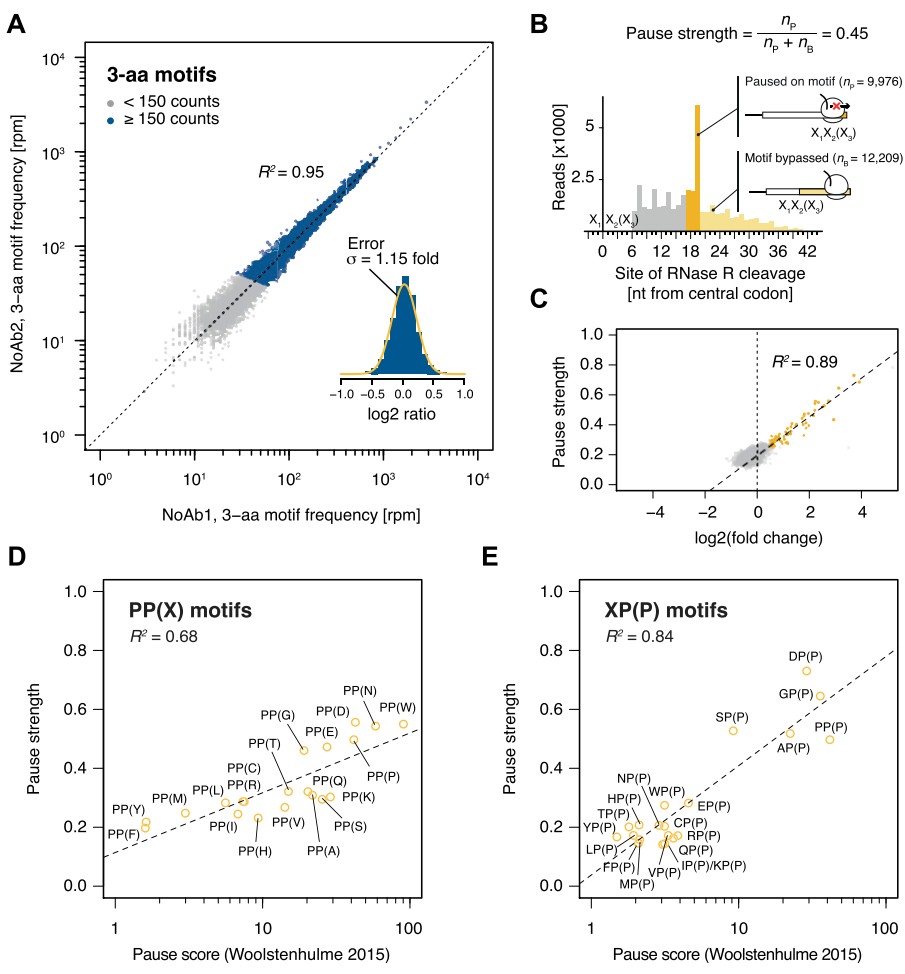

**Figure 2. Motif pause strength correlates with enrichment on inverse toeprinting.**
**(A)** 3-aa motif frequencies in reads per million (rpm) from two independent inverse toeprinting experiments performed after translation in the absence of antibiotic (NoAb1 and NoAb2). The inset represents a histogram of log2 ratios between replicates for 3-aa motifs having low statistical counting error (i.e., with >150 counts [blue], Fig S5), with an overlaid normal error curve (mean = 0.02, SD = 0.2 log2 units, equivalent to $\sigma$ = 1.15 fold). **(B)** Formula used to calculate pause strength for an $X_1X_2(X_3)$ motif, with the amino acid in the ribosomal A-site in brackets. **(C)** Plot of pause strength against log2 (fold change) of all possible 3-aa motif frequencies relative to the NNS$_{15}$ library. Yellow points correspond to intrinsic 3-aa pause motifs with a pause strength ≥0.25. All other motifs are shown as gray dots. **(D, E)** Plot of pause strengths calculated in this study against pause scores calculated from ribosome profiling data obtained from *E. coli* cells lacking EF-P, for (D) PP(X) and (E) XP(P) motifs (Woolstenhulme et al, 2015). The scores obtained by both methods are strongly correlated, as indicated by $R^2$ values of 0.68 and 0.84 for PP(X) and XP(P) motifs, respectively.

motifs were well represented in the input library (i.e., only 11 motifs had <150 reads in the sequenced NNS$_{15}$ library), poorly measured 3-aa motifs were depleted during the selection process because they did not induce sufficiently strong pauses in translation. Their exclusion therefore does not limit or bias our analysis.

Next, we devised a means to quantify the strength of translational pausing for each 3-aa motif, which we call the pause strength (Fig 2B). This represents the number of times a ribosome is stalled on a given motif divided by the number of times ribosomes have encountered that motif. Consequently, a value of one indicates a very strong pause (i.e., ribosomes never bypass that motif), whereas a value of 0.2 indicates a weak pause (i.e., 80% of the ribosomes that encountered this motif bypassed it and stalled at another motif downstream on the mRNA). We found a strong linear correlation between the pause strength and the log2 (fold change) in frequency for all motifs exhibiting a pause strength greater than 0.25 (Fig 2C). By contrast, the change in 3-aa motif frequency on selection did not correlate with *E. coli* codon usage frequencies, indicating that translational pauses induced by consecutive rare or low usage codons are not detected by inverse toeprinting (Fig S6). Pause strengths were in the range of 0.1–0.8, with values less than 0.25 found for most motifs. In the absence of antibiotic, PP(X) and XP(P) motifs displayed pause strengths that were strongly correlated

with the "pause scores" calculated for the same motifs from ribosome profiling experiments performed in EF-P-deficient *E. coli* (Woolstenhulme et al, 2015) (Fig 2D and E). Although PP(X) motifs on the whole appear to be more efficient at pausing translation (PP(D) > PP(W) > PP(N) > PP(P) > PP(E) > PP(G)), the strongest XP(P) motifs (DP(P) > GP(P) > SP(P) > AP(P) > PP(P)) exhibit the greatest pause strengths. We also identified an additional intrinsic arrest motif (XP(C)), where X matches the amino acids in XP(P) motifs that cause the strongest pauses (i.e., A, D, G, S).

We then sought to identify short 3-aa motifs that arrest ribosomes in response to the macrolide antibiotic Ery. Previous studies have shown that ribosomes translating in the presence of Ery stall when they encounter +X(+) motifs ("+" being the positively charged amino acids arginine or lysine) (Davis et al, 2014; Kannan et al, 2014; Sothiselvam et al, 2014). We performed inverse toeprinting on an in vitro translation reaction using the NNS$_{15}$ library in the presence of Ery. Paired-end Illumina sequencing of the corresponding inverse toeprints revealed a strong tri-nucleotide periodicity of fragment sizes compared with the samples without antibiotic (Fig S7). Comparison of independent biological replicates, obtained in the presence of Ery, allowed us to identify a subset of well-measured 3-aa motifs derived from inverse toeprints (Fig S8). From this subset, we measured the enrichment of 3-aa motifs in

inverse toeprints obtained after translation in the presence of Ery relative to those obtained in the absence of antibiotic, and calculated a mean error of 1.2-fold change in motif frequency upon addition of Ery (Fig 3A). Several classes of 3-aa motifs that were significantly enriched in the Ery sample were characterized by relatively high pause strengths (Figs 3A and C, S9, S10), not seen in the absence of the drug (Fig 3B). These include the previously reported +X(+) motif, the general XP(X) motif and its subsets +X(W) and XP(W), with the latter's ability to induce pausing in vitro correlating well with the results of our in vivo assay (Fig 3D and E). Among these motifs, most XP(X) motifs induced significant drug-dependent pauses in the presence of EF-P, but not in its absence (Fig S11). The molecular basis for this phenomenon is unclear at present and will need to be further investigated.

Thus, inverse toeprinting provided us with a detailed view of the translational pausing landscape of drug-free and drug-bound *E. coli* ribosomes that matches and complements earlier in vivo profiling data (Woolstenhulme et al, 2015). Motif enrichment upon inverse toeprinting and pause strength are strongly correlated, indicating the robustness of the selection procedure. This will be useful in identifying longer and hence less frequently occurring arrest motifs for which pause strengths cannot be calculated.

## Inverse toeprinting identifies arrest peptide variants that discriminate between closely related ligands

To test the ability of inverse toeprinting to detect targets in a high complexity library, we used the method to identify ErmBL variants (Fig S2B) that were differentially enriched in the presence of Ery (Fig 4A) or of the weaker antibiotic oleandomycin (Ole) (Fig 4B). After paired-end Illumina sequencing, ~1.3 million library reads were aligned to the wild-type *ermBL* sequence, corresponding to 724,573 unique protein variants. The observed distribution of mutations (100% of single mutants and ~55% of double mutants) within the library (Table S1) closely approximated the expected distribution based on the mutation rate of 7% used at each nucleotide position of *ermBL*. Inverse toeprints obtained after translation in the presence of Ery or Ole featured >230,000 unique protein variants in each case, ~80% of which were sequenced more than once.

Comparison of the sequence variants that were obtained in the presence of Ery or Ole revealed amino acids that were over- or underrepresented at each position of ErmBL in response to the antibiotics (Fig S12). Leucine was enriched at position seven of ErmBL (−3 relative to the amino acid in the ribosomal P-site) in the presence of Ery, but not Ole. Several mutants encoding leucine at this position underwent translational arrest exclusively in the presence of Ery (Table S2). We further characterized the L7 single mutant and L7K8 double mutant of ErmBL, both of which were enriched ~1.9-fold in the Ery sample compared with the input library, whereas they were not significantly enriched in the Ole sample (0.7-fold and 0.5-fold, respectively). The L7 and L7K8 mutants originated from 198 and 55 unique variants at the nucleotide level, respectively, indicating multiple independent events. Using an in vitro toeprinting assay (Fig 4C) and an in vivo reporter to measure translational arrest (Bailey et al, 2008) (Fig 4D), we confirmed the antibiotic specificity for the L7 and L7K8 mutants. The weak Ery-dependent toeprints obtained in vitro contrasted with

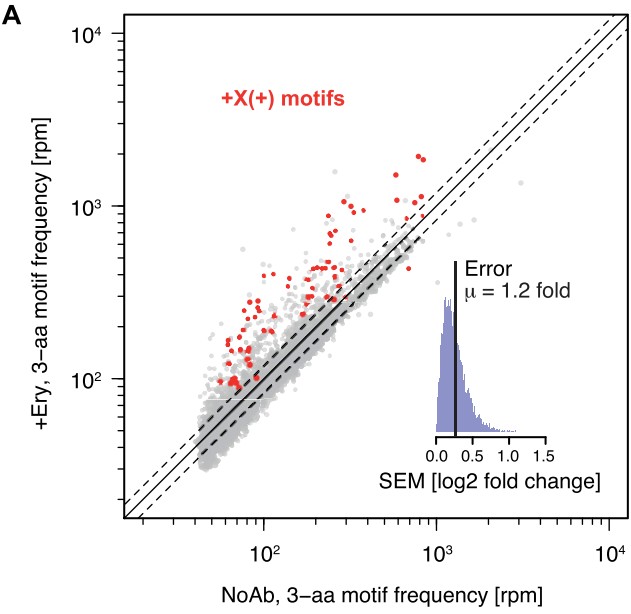

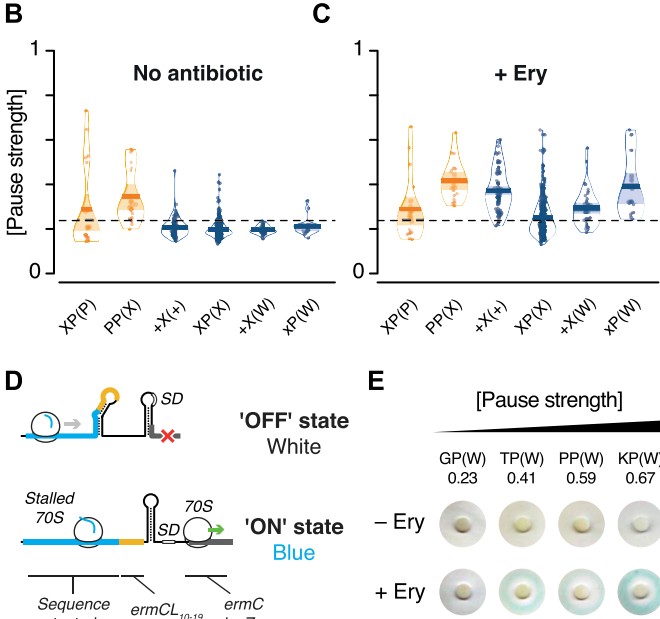

**Figure 3. Nascent peptide-dependent translational arrest in response to Ery.**
**(A)** Frequency of occurrence of 3-aa motifs with low statistical counting error in inverse toeprints obtained in the absence or presence of Ery, with +X(+) motifs indicated in red. The inset represents a histogram of the SEM of the log2 fold change in 3-aa motif frequency upon addition of Ery. The upper and lower dotted lines (gray) indicate 1.20 and 0.83-fold changes, respectively, corresponding to the mean ($\mu$) of the distribution of SEM (log2 fold change). **(B, C)** RDI (raw data, description and inference) plot showing pause strengths for individual motifs translated in the (B) absence or (C) presence of Ery. Polyproline motifs are shown in yellow and all other motifs are in blue. The horizontal dashed line corresponds to the 0.25 pause strength cutoff used to identify motifs that are enriched upon addition of Ery. **(D)** Overview of the *lacZα*-complementation assay used to test the in vivo activity of ErmBL variants (modified from Bailey et al [2008]). **(E)** Disc-diffusion test plates used to assay the ability of nascent formyl-MAXP(W) to cause translational arrest in vivo. Discs marked with +Ery contain this antibiotic and blue rings result from the induction of a *lacZα* reporter in response to ribosome stalling at an upstream test ORF (modified from Bailey et al [2008]).

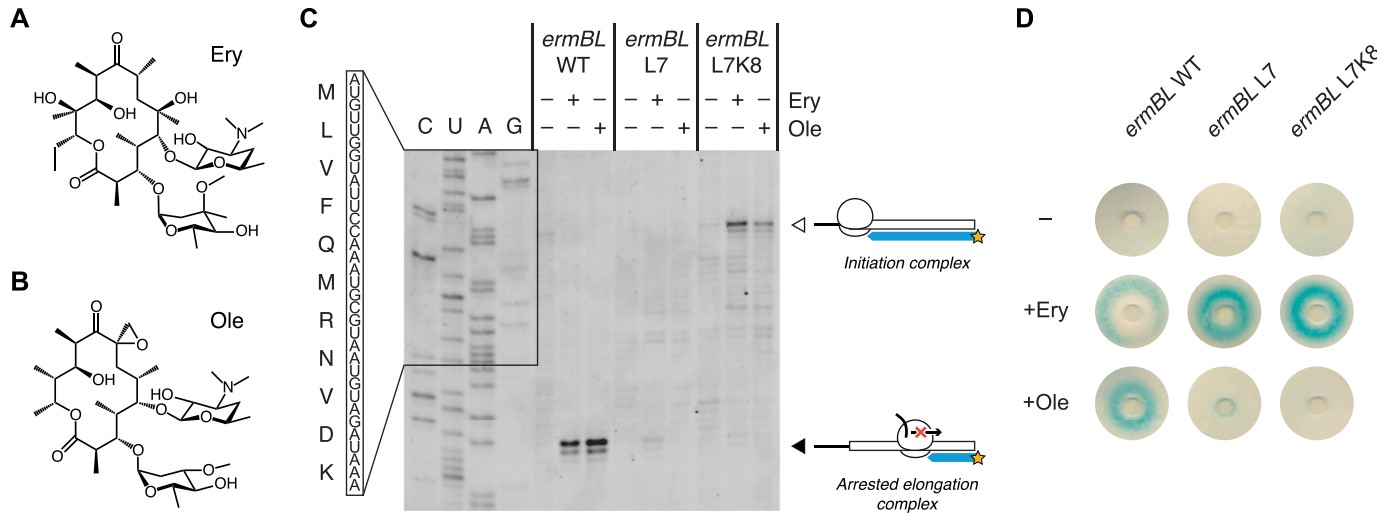

**Figure 4. The ErmBL L7 and L7K8 mutants discriminate between closely related antibiotics.**
**(A, B)** Chemical diagrams for Ery and Ole. **(C)** Classical toeprinting analysis of translational arrest by wild-type ErmBL (*ermBL* WT), an L7 single mutant (*ermBL* L7), and an L7K8 double mutant (*ermBL* L7K8), in the absence or presence of the antibiotics Ery and Ole. The white arrow indicates ribosomes on the start codon, and the black arrow indicates arrested elongation complexes with the GAU codon encoding Asp-10 in the ribosomal P-site. The sequence of wild-type ErmBL is shown. **(D)** Disc-diffusion test plates used to assay the ability of nascent ErmBL WT, ErmBL L7, and ErmBL L7K8 to cause translational arrest in vivo in the absence or presence of Ery or Ole soaked into paper discs. It should be noted that the double mutant shows greater antibiotic selectivity in vivo compared with the single mutant, as indicated by the light blue ring observed in the +Ole condition for the ErmBL L7 mutant.

strong β-galactosidase activity in vivo, indicating that arrest sequences that appear weak by toeprinting can stall ribosomes effectively in vivo. The L7 and L7K8 mutants could not have been predicted based on the available structure of a stalled ErmBL-70S complex (Arenz et al, 2016) and would not have been identified using alanine-scanning mutagenesis of ErmBL. The weak toeprint signals would also likely have been overlooked, highlighting the value of inverse toeprinting for exploring the sequence space of known arrest peptides or for identifying specific variants out of a high complexity library.

## Discussion

We developed inverse toeprinting, an in vitro selection method that locates stalled ribosomes on the mRNA with codon resolution while also preserving the coding region upstream of the point of arrest. This presents a major advantage over current methods, such as ribosome profiling, when a reference genome is not available for mapping protected mRNA footprints, but the encoded amino acid sequence needs to be known. The complexity of the input transcript library can be fine-tuned to match the biological question to be addressed, making it possible to limit the reaction volume and measure the effect of tens if not hundreds of different conditions in a parallel fashion. Importantly, inverse toeprinting can provide comparable results with in vivo methods such as ribosome profiling, making it a robust and convenient framework for developing more complex methods.

Here, we used inverse toeprinting to characterize the translational pausing landscape of free and drug-bound bacterial ribosomes. The large number of conditions that can be analyzed at

a time with this technique makes it a valuable tool for the rapid characterization of antibiotics and compounds that inhibit the bacterial ribosome. For example, sequence-dependent inhibition profiles could be established for entire collections of macrolide derivatives obtained by a novel total synthesis approach (Seiple et al, 2016; Dinos, 2017). This would, in turn, provide fresh insights into the mechanism of macrolide action that could inform the development of new drugs by rational design. Another application of inverse toeprinting is the identification of arrest peptide variants with specificities for closely related small molecules, as shown by the ErmBL variants that respond differently to chemically similar antibiotics. A similar strategy could be applied to naturally occurring arrest peptides such as TnaC (Gong et al, 2001) or the fungal arginine attenuator peptide (Luo & Sachs, 1996), which arrest translation in response to elevated levels of amino acids in the cell. Biological sensors that respond to unnatural amino acid derivatives or other compounds of interest in a dose-dependent manner could find uses in a number of synthetic biology or biotechnological applications.

Identifying small molecule-dependent arrest peptides encoded within a truly random transcript library poses some additional challenges. Indeed, these peptides are likely to occur very infrequently and may therefore be more difficult to isolate that the much larger number of functional variants that exist within a focused library. Consequently, it may be necessary to incorporate inverse toeprinting into a more complex and more sensitive SELEX-like procedure. We have shown that the enrichment of arrest-inducing sequences is strongly correlated with their pause strength. This is important as it establishes the effectiveness of inverse toeprinting as a selection tool to identify rare arrest sequences hidden within a high complexity library. Although our method allows us to perform consecutive rounds of selection, it will

be necessary to develop a counter-selection procedure to reduce the number of false positives after each round. In particular, we will need to address the incomplete recycling of ribosomes at stop codons that is observed currently.

The fact that inverse toeprinting is an in vitro method offers some advantages relative to in vivo approaches when it comes to identifying arrest peptides that sense small molecules. Indeed, problems may arise in vivo because of the inefficient uptake of small molecules into the cell, their degradation, modification, or their accelerated efflux out of the cell. An additional advantage of our in vitro method is that it enables us to dissect molecular processes in isolation by giving us direct control over the reaction conditions. Being able to remove certain molecular processes from their cellular context is a double-edged sword however, and care must be taken to validate results obtained by inverse toeprinting with in vivo assays. For the systems studied here, the correlation between in vitro and in vivo data is generally good, but the small discrepancies observed could be due to the NNS$_{15}$ library's intrinsic focus on the early cycles of translation, where the nascent peptide is not yet fully threaded inside the ribosomal exit tunnel. These and other as yet unforeseen considerations will need to be taken into account as the method is developed further.

# Materials and Methods

## Method overview

The DNA template encoding a T7 RNA promoter, followed by a ribosome binding site, a potential arrest peptide, a fixed "spacer" region of four codons, two TGA stop codons, and an *EcoRV* restriction site are generated by PCR and subsequently transcribed in vitro using T7 RNA polymerase with an excess of thio-phosphate-GMP, which can only be incorporated at the 5' end of the mRNA. In the next step biotin-maleimide is coupled to the 5' thiol group on the mRNA. The 3' polyA-tail needed for efficient degradation by RNase R is added using Poly-A polymerase. ~5 pmol of this 5'-biotinylated and 3'-polyadenylated mRNA is then used as a template for in vitro translation using a PURExpress kit (NEB) from which RF-2 is omitted to prevent the release of ribosomes that translate beyond the spacer sequence. Using an NNS (a**N**y a**N**y **S**trong—i.e., G or C) library ensures that no other UGA stop codons appear in the variable region. By contrast, ribosomes that reach the UAG stop codons found within the variable region can be released using RF-1.

Ribosomes engaged in translation of the mRNAs can either stall on the variable coding region if it encodes an arrest peptide or can translate until they reach the stop codon downstream of the spacer. RNase R is then used to degrade mRNAs from their 3' end in the presence of 50 mM Mg$^{2+}$, which inactivates and stabilizes ribosomes on the mRNA, thus ensuring that the upstream coding region is protected. Transcripts without ribosomes are degraded in this step. After this step, mRNAs are subjected to a phenol-chloroform extraction to ensure the inactivation and removal of RNase R and of the ribosomes. The mRNAs are purified via their 5'-biotin using streptavidin-coupled Dynabeads. In the next step,

a DNA oligonucleotide linker is attached enzymatically to the 3' end of the mRNA. This linker contains a fixed "spacer" region, followed by three TGA stop codons, one in each reading frame, followed by a restriction site. Two different linker oligonucleotides were used in this study, one encoding an *ApoI* restriction site and one encoding an *EcoRV* restriction site. Adding the linker generates a 3' end of known sequence needed for reverse transcription of the mRNA. After second strand synthesis, the double stranded cDNA is treated with the restriction enzyme encoded in the DNA template (*EcoRV* for odd rounds of selection and *ApoI* for even rounds). Ribosomes that reach the stop codon (and thus translated a sequence that does not induce stalling) protect the restriction site from RNase R degradation, thus allowing restriction enzymes to cut these DNAs and prevent their amplification in the following PCR step. Double stranded cDNAs derived from mRNAs coding for peptides that arrested the ribosome do not contain the restriction site and are consequently amplified in the ensuing PCR. The *EcoRV* and *ApoI* sites have a stretch of at least four A/Ts and thus cannot occur within the NNS$_{15}$ region. PCR products from this step serve as templates for either another round of inverse toeprinting through the addition of a T7 promoter, or for NGS library generation and deep sequencing.

## Experimental procedures

DNA and RNA products at various points in the inverse toeprinting protocol were analyzed on 9% acrylamide (19:1) TBE (90 mM Tris, 90 mM boric acid, and 2 mM EDTA) gels and stained with SyBR Gold (Invitrogen). Inverse toeprints were excised from 12% acrylamide TBE gels using a clean scalpel. Gels to analyze RNA were run under denaturing conditions (8 M urea in the gel). All reactions were performed using molecular biology grade H$_2$O (Millipore). Oligonucleotides used in this study are listed in Tables S3–S6.

### DNA template generation
All DNA templates for inverse toeprinting were generated by PCR with Phusion DNA polymerase, using an oligonucleotide encoding the variable region in combination with oligonucleotides *T7_RBS_ATG_f* and *Stop_EcoRV_r* as templates. Amplification was performed using oligonucleotides *T7_f* and *EcoRV_r* in 10-fold excess. PCR conditions included an annealing temperature of 64°C and 20 cycles of amplification. DNA templates were generated using a total of 5 pmol *ermBL_deep_mutated* oligonucleotide or 10 pmol *NNS15* oligonucleotide, respectively. PCR products were purified using a PCR purification kit (Qiagen) according to the manufacturer's instructions and were used as templates for in vitro transcription.

### In vitro transcription
The DNA template encodes a T7 promoter followed by an optimized Shine-Dalgarno sequence, according to the instructions of the NEB PURExpress system handbook. In vitro transcription was performed using T7 RNA polymerase (Promega) in a buffer containing 80 mM Tris–HCl, 24 mM MgCl$_2$, 2 mM spermidine, and 40 mM DTT, pH 7.6, in the presence of 7.5 mM ATP (Sigma Aldrich), CTP and UTP, 0.75 mM GTP (CTP, UTP, and GTP from Jena Bioscience), and 6.75 mM Thio-Phosphate-GMP (Genaxxon). In the first round of inverse toeprinting, 10 ng/µl of DNA template were used; for further rounds,

the amount was reduced to 1 ng/μl. In vitro transcription was performed at 37°C for 2–3 h, mRNA was purified by phenol-chloroform extraction, washed three times with chloroform and precipitated using 0.1 volumes of $NH_4$-acetate (10 M) and 1 volume of isopropanol. To remove unincorporated nucleotides, the recovered mRNA was subsequently washed through Amicon membrane centrifugal concentrators with a molecular weight cutoff (MWCO) of 30 kD (Millipore) until the flow-through was free of unincorporated nucleotides (as determined by NanoDrop measurements). The final concentration of mRNA was determined using the NanoDrop.

### Biotinylation

Biotinylation was performed using a 1,000-fold excess of biotin-maleimide (Vectorlabs) over mRNA 5′ ends. According to the manufacturer's instructions, the biotin-maleimide was dissolved in dimethylformamide. 600 pmol mRNA were mixed with 600 nmol biotin-maleimide in 100 mM in Bis-Tris-acetate buffer pH 6.7 and incubated at room temperature for 3 h. Unincorporated biotin was removed by washing the mRNA three times with $H_2O$ (molecular biology grade, Millipore) in an Amicon membrane centrifugal concentrator with a MWCO of 30 kD (Millipore). mRNA was recovered and biotinylation efficiency was analyzed using a dot blot.

### Dot blot

$H^+$ bond membrane (GE Healthcare) was treated with 6× SSC buffer (900 mM NaCl, 90 mM $Na_3$-citrate, pH 7.0) for 10 min and dried briefly between two pieces of Whatman paper. Samples and the standard were diluted in 6× SSC buffer to 0.5, 1.0, 2.5, and 5.0 μM, and 1 μl of each dilution was pipetted onto the prepared membrane. The membrane was then baked for 2 h at 80°C to attach the mRNA to the membrane. The membrane was subsequently blocked in 2.5% dry milk solution in TBS-T (50 mM Tris–HCl, 150 mM NaCl, and 0.05% [vol/vol] Tween-20, pH 7.5) for 1 h at room temperature. The milk solution was removed and the membrane was incubated with streptavidin-alkaline phosphatase antibody (Promega) in a 1:1,000 dilution in TBS-T for 1 h at room temperature. Unbound antibody was removed by washing three times with TBS-T buffer. Detection was performed using the NBT/BCIP detection kit (Promega) in alkaline phosphatase buffer according to the manufacturer's instructions. The detection reaction was stopped by two washes with TBS-T buffer and an image of the membrane was acquired immediately on a Bio-Rad Imager. The biotinylation efficiency was estimated from the intensity of the sample dots compared with the intensity of the standard dots.

### Poly-adenylation of the mRNA 3′ end

Poly-adenylation of the biotinylated mRNA was performed using Poly-A polymerase (NEB) using the supplemented buffer. The ratio of mRNA 3′ ends to ATP molecules was chosen to be 1:100. The reaction was incubated at 37°C for 2–3 h and poly-adenylation efficiency was assessed by denaturing PAGE (9%). Polyadenylated mRNA was subjected to a phenol-chloroform extraction, washed three times with chloroform, and precipitated with $NH_4$-acetate-isopropanol with 0.5 μl GlycoBlue (Thermo Fisher Scientific). Unincorporated ATP was removed by successive washes in Amicon

membrane centrifugal concentrators with a MWCO of 30 kD (Millipore).

### RNase R activity

Purified RNase R (Suzuki et al, 2006) was provided by Dr. Arun Malhotra (University of Miami) and was used at 1 mg/ml stock solution. 5 pmol of mRNA were used to test the degradation efficiency of 2 μl RNase R on every batch of mRNA in a buffer containing 50 mM Hepes-KOH, 100 mM K-glutamate, 50 mM Mg-acetate, and 1 mM DTT, pH 7.5. Time points were taken directly into RNA loading dye (95% formamide, 250 μM EDTA, 0.25% [wt/vol] bromophenol blue, and 0.25% [wt/vol] xylene cyanol) before addition of the enzyme and after 5, 10, and 30 min of incubation at 37°C. The samples were analyzed by denaturing PAGE (9%), stained with SyBR Gold to monitor mRNA degradation.

### Inverse toeprinting

PURExpress Δ RF-123 kit (NEB) was used to perform in vitro translation. ~5 pmol of 5′-biotinylated and 3′-polyadenylated mRNA were used as a template. Antibiotics (Ery, Ole) were supplemented at a final concentration of 50 μM in 10 μl reactions. RF-1 and RF-3 were added to the translation reaction according to the manufacturer's instructions. Translation was performed at 37°C for 30 min before the samples were placed on ice and 10 μl ice-cold $Mg^{2+}$ buffer (50 mM Hepes-KOH, 100 mM K-glutamate, 87 mM Mg-acetate, and 1 mM DTT, pH 7.5) was added to the reactions, thus increasing the $Mg^{2+}$ concentration to 50 mM. 2 μl of RNase R (1 mg/ml) was added, followed by an additional incubation for 30 min at 37°C for RNase R-mediated mRNA degradation. Ribosome-protected mRNA was purified by phenol-chloroform extraction, washed three times with chloroform, and precipitated using $NH_4$-acetate-isopropanol. RNA was recovered by centrifugation at full speed for 30 min at 4°C and resuspended in 50 μl 1× BWT buffer (5 mM Tris–HCl, 0.5 mM EDTA, 1 M NaCl, and 0.05% [vol/vol] Tween-20, pH 7.5).

### mRNA purification with dynabeads

5 μl M-280 streptavidin Dynabeads (Life Technologies) were prepared for each sample by washing three times with 1× BWT buffer in DNA loBind tubes (Eppendorf) and resuspended in 50 μl of the same buffer. Dynabeads and purified RNA from the previous step were combined in these tubes and incubated on a tube rotator for 15 min at room temperature to allow binding of the biotinylated mRNA to the streptavidin beads. After incubation, the beads were collected using a magnet and the supernatant was discarded. The beads were washed two times with 1× BWT buffer to remove unincorporated RNA, followed by two washes with $H_2O$ to remove the 1× BWT buffer. Beads were resuspended in 4.5 μl $H_2O$.

### Linker ligation

The beads from the previous step were combined with 10 pmol (1 μl) of the desired linker (3′_linker_ApoI or 3′_linker_EcoRV depending of the round of selection) plus 3 μl PEG 8,000, 1 μl PNK buffer, and 0.5 μl T4 RNA ligase 2, truncated (all NEB). Incubation was performed on a tube rotator for 2 h at room temperature. After incubation, the beads were washed three times with $H_2O$ to remove unincorporated linker oligonucleotide and were resuspended in 12 μl $H_2O$.

### Reverse transcription

12 µl beads were combined with 1 µl *Linker_r* oligonucleotide (2 µM) and 1 µl 2'-deoxynucleotide triphosphates (dNTPs) (NEB, 10 mM per dNTP), and incubated at 65°C for 5 min to allow annealing of the *Linker_r* oligonucleotide to the linker. After annealing, 4 µl of first strand buffer, 1 µl of 100 mM DTT, and 1 µl of superscript III (all Invitrogen) were added and the reaction incubated at 55° for 30 min to allow reverse transcription of the Dynabead-bound mRNA.

### PCR on cDNA, restriction digestion

Reverse transcribed cDNA was used without further purification as a template for PCR. To generate double stranded DNA for restriction digestion, a fill-up reaction was performed using *cDNA_f* oligonucleotide and the reverse transcribed cDNA (10 s denaturation, 5 s annealing at 42°C, and 30 s elongation at 72°C). The resulting dsDNA was combined with 1 µl of the respective restriction enzyme and the sample was incubated at 37°C for 1 h. To amplify undigested DNA, *Linker_r* oligonucleotide was added and a PCR performed with 8–14 cycles (denaturation at 98°C for 10 s, annealing at 42°C for 5 s, and elongation at 72°C for 10 s). The number of PCR cycles giving the best results was used for further purification of the cDNA.

### Purification of DNA fragments of interest after PCR

Wild-type and *ermBL* samples were purified using a homemade electro-elution device. PCR products were analyzed by TBE-PAGE (12%) and bands of interest were excised from the gel using a clean scalpel. Gel pieces were crushed through a 5 ml syringe into 50 ml Falcon tubes whose base (1–2 ml) had been cut off and covered with Parafilm. The crushed gel pieces were then embedded into a new 9% acrylamide TBE gel inside the cut Falcon tube (~8 ml of gel solution). After polymerization, DNA was eluted from the gel by filling the Falcon tube with TBE buffer (upper buffer reservoir) and hanging the Falcon tube into a beaker filled with TBE buffer (lower buffer reservoir, ice-cooled). DNA was eluted into the lower buffer reservoir by placing clean electrodes into the two buffer reservoirs and by applying a current of 10 W per gel (two gels max) for 30 min. The buffer from the lower reservoir was recovered and the eluted DNA was concentrated using Amicon membrane centrifugal concentrators with a MWCO of 30 kD (Millipore). DNA was precipitated by addition of $NH_4$-acetate isopropanol and 0.5 µl GlycoBlue (Thermo Fisher Scientific) and incubation at –20°C for 1 h.

The $NNS_{15}$ library samples were purified as described, with modifications (Ingolia et al, 2012). After cutting out the bands of interest, the gel pieces were crushed through a 5 ml syringe into 15 ml Falcon tubes and 7.5 ml of gel elution buffer (10 mM Tris–HCl, pH 8.0, 500 mM Na-acetate, and 0.5 mM Na-EDTA) were added. The tubes were incubated on a tube rotator at room temperature overnight. Gel debris was separated from the buffer by filtering through 0.22 µm centrifugal filters (Millipore). Each sample was then concentrated to ~250 µl using a SpeedVac. The eluted DNA was precipitated using 0.1 volume $NH_4$-acetate and 2.5 volumes ethanol with 0.75 µl GlycoBlue (Thermo Fisher Scientific) and incubation at –20°C for 1 h.

After precipitation, DNA was recovered by centrifugation in a tabletop centrifuge at full speed for 30 min at 4°C. The supernatant was carefully removed by pipetting and the DNA pellet briefly dried in the SpeedVac for 10–15 min. The cDNA was then resuspended in 15 µl $H_2O$ (molecular biology grade) and used as a PCR template for the addition of the T7 promoter sequence or the NGS adapters.

### Addition of the T7 promoter for another round of inverse toeprinting

The purified cDNA from the previous step was used as a template for PCR in combination with *T7_RBS_ATG_f* (1 µM), encoding the T7 promoter needed for in vitro transcription, *T7_f* and *Linker_r* oligonucleotides (10 µM each). 8–14 cycles of PCR (64°C annealing temperature) were used and amplified DNA was purified using a Qiagen PCR purification kit. The concentration of purified DNA was determined using a NanoDrop.

### Addition of NGS adaptor to purified cDNA

Long NGS adaptor oligonucleotides contain the Illumina TruSeq adapter sequences followed by 18 nucleotides complementary to the 5' or 3' region of the cDNA. The reverse NGS oligonucleotides also encode barcode sequences for multiplexing according to the TruSeq v1/v2/LT protocol (Illumina). The adaptors were added to the cDNA by PCR (8–14 cycles) using the long oligonucleotides (20–26) in 1 µM stock solutions and the short amplifying oligos (18, 19) in 100 µM stock concentration. PCR products were purified using a Qiagen PCR purification kit. The size and concentration of the fragments obtained were analyzed using a 2100 Agilent Bioanalyzer with the DNA 1000 kit.

### Preparation of mRNA template library for NGS

To prepare the mRNA library for NGS 5 pmol of mRNA was reverse transcribed using Superscript III (Invitrogen) using an NGS adapter (27/28) containing a stretch of 18 nucleotides reverse complementary to the fixed "spacer" region in the 3' end of the mRNA. The resulting cDNA served as template for subsequent PCR using the forward (20) and reverse adapter (27/28) in 1 µM stock concentration and the short amplifying oligonucleotides in 100 µM stock concentration. PCR products were purified using a Qiagen PCR purification kit. The size and concentration of the fragments obtained were analyzed using a 2100 Agilent Bioanalyzer with the DNA 1000 kit.

### Next generation sequencing

Next generation sequencing was performed by the Tufts Genomics Core Facility in Boston, USA on an Illumina HiSeq 2500 system in rapid run mode with 150 PE read.

### Toeprinting

Toeprinting to test novel arrest sequence motifs was performed using the PURExpress ΔRF-1,2,3 kit (NEB). DNA templates were generated by PCR using *T7_RBS_ATG_f*, *TP_3'_spacer_r* and *TP_NV1_r* oligonucleotides (all at 1 µM) in combination with the oligonucleotide encoding the sequence to be tested. These oligonucleotides served as templates and were amplified using the *T7_f* and *TP_NV1_r_short* oligonucleotides (100 µM) with Phusion DNA polymerase. PCR products were purified using the Qiagen PCR purification kit and eluted with $H_2O$ (molecular biology grade). Ery or Ole were dried into the tube to yield a final concentration of 50 µM in the 5 µl toeprinting reaction. 1 pmol of DNA template was

combined with 2 μl of solution A and 1.5 μl solution B of the PURExpress system. The reaction was incubated at 37°C for 15 min before addition of 1 μl of the 5′-Yakima Yellow labeled NV1 probe (Vázquez-Laslop et al, 2008) (2 μM) and the reaction was incubated for another 5 min at 37°C. Reverse transcription was performed with 0.1 μl dNTPs (10 mM stock of each dNTP), 0.4 μl PURE system buffer, 0.5 μl AMV RT (Promega), and 20 min incubation at 37°C. After generation of the Yakima Yellow-labeled cDNA, the mRNA was degraded by addition of 0.5 μl 10 M NaOH and incubation at 37°C for 20 min. The samples were neutralized with 0.7 μl 7.5 M HCl. 20 μl toeprint resuspension buffer (300 mM Na-acetate pH 5.5, 5 mM EDTA, and 0.5% SDS) and 200 μl PNI buffer were added to each sample and cDNA was purified using the Qiagen nucleotide removal kit. The cDNA was eluted using 50 μl of $H_2O$ (molecular biology grade). cDNA was dried into the tube using a SpeedVac and resuspended in 6 μl toeprint loading dye (95% formamide, 250 μM EDTA, and 0.25% bromophenol blue). Samples were denatured at 95°C for 5 min before loading onto a 7.5% polyacrylamide TBE sequencing gel containing 8 M urea. The gel was run at 40 W and 2,000 V for 2.5 h. Yakima-Yellow labeled cDNAs were detected using a Typhoon Gel Scanner in fluorescent mode.

### Disc diffusion assay

We used the LacZα-based in vivo system described by Bailey et al (2008). Using oligonucleotides 39–50, we generated by PCR several plasmids in which we replaced the first 10 codons of the encoded *ermCL* with the sequence of interest, thus maintaining the regulatory region of *ermCL* and *ermC*. We transformed these plasmids into chemically competent *E. coli* TB1 cells. Transformants were grown for 6 h in LB media with 1 mM IPTG at 37°C to an $OD_{600}$ of 1.5. For the MAXPW motifs, 50 μl of IPTG (100 mM) were added to LB-agar plates containing ampicillin (100 μg/ml) and streptomycin (50 μg/ml) before the addition of 100 μl cells. Whatman filter discs soaked with 10 μl 5-bromo-4-chloro-3-indolyl-β-ᴅ-galactopyranoside (X-gal) (50 μM) and either water or 50 μg Ery were placed onto the agar plates. For the ErmBL constructs, 50 μl of IPTG (100 mM) and 50 μl of X-gal (50 μM) were added to LB-agar plates containing ampicillin (100 μg/ml) and streptomycin (50 μg/ml) before addition of 100 μl cells. Whatman filter discs soaked with water or 50 μg Ery or Ole were placed onto the agar plates. Agar plates were incubated at 37°C for 24 h and images were acquired immediately after the incubation period.

### Expression and purification of EF-P

Expression of EF-P was performed together with the expression of the modification enzymes as described previously (Ude et al, 2013) in *E. coli* BL21 Gold, and expression was induced using 1 mM IPTG at an $OD_{600}$ of 0.6. Harvested cells were resuspended in lysis buffer (50 mM Hepes-KOH, pH 7.6, 10 mM $MgCl_2$, and 1 M $NH_4Cl$) and sonicated. Cell debris was removed by centrifugation (45 min, 40,000 *g*, 4°C) and the clarified lysate was mixed with cobalt-agarose (Sigma-Aldrich) and incubated on a tube rotator at 4°C for 1 h. The cobalt-agarose was washed with lysis buffer and the protein was eluted with lysis buffer containing 250 mM imidazole. The eluate was concentrated using centrifugal concentrators with a MWCO of 10 kD (Millipore). To exchange the buffer to protein storage buffer (20 mM Hepes-KOH, pH 7.6, 10 mM $MgCl_2$, 50 mM KCl,

and 50 mM $NH_4Cl$), gel filtration was performed through a Superdex 75 matrix using an NGC medium pressure liquid chromatography system (Bio-Rad). Eluate fractions were analyzed by SDS–PAGE and protein-containing fractions were pooled and concentrated to a final concentration of 20 mg/ml using centrifugal concentrators with a MWCO of 10 kD (Millipore). The activity of the purified EF-P was assessed by toeprinting using a DNA template encoding the sequence MMHHHHHHRPPPI. Addition of EF-P to a final concentration of 10 μM efficiently rescued ribosomes stalled on the poly-proline motif.

### Data analysis

Unless it is indicated otherwise, data analysis was carried out using a series of custom scripts written in-house in Python, which relied on the use of the Biopython package (Cock et al, 2009).

### Read assembly and trimming

Read pairs were assembled using PEAR v0.9.10 (Zhang et al, 2014) on a Mac Book Pro with a 2.7 GHz Intel Core i7 processor and 16 GB 1,600 MHz DDR3 memory, with the maximal proportion of uncalled bases in a read set to 0 (−u option) and the upper bound for the resulting quality score set to 126 (−c option).

Regions immediately upstream of the start codon and downstream of the point of cleavage by RNase R were removed using a modified version of the *adaptor_trim.py* script written by Brad Chapman (https://github.com/chapmanb/bcbb/blob/master/align/adaptor_trim.py). The 5′ flanking region was defined as GTATAAGGAGGAAAAAT, whereas the 3′ flanking region was GCGATCTCGGTGTGATG for the $NNS_{15}$ and ErmBL libraries, and GGTATCTCGGTGTGACTG for all other samples. A maximum of two mismatches within each of these flanking regions was tolerated, whereas all other reads were discarded. Trimming of the retained reads resulted in sequences with a start codon directly at the 5′ end and, in the case of samples resulting from inverse toeprinting, the site of RNase R cleavage at the 3′ end.

### Quality filtering and selection of the region of interest

Trimmed sequences were further processed with our *process-reads.py* script. Reads featuring ≥18 "A" bases within the first 22 nucleotides from the start codon were eliminated, as were reads from which the expected "ATG" start codon was absent. Sequences within the region of interest were retained for further analysis, provided that each base call within this region had a Q score of 30 or more. For the $NNS_{15}$ library sample, the region of interest spanned nucleotides 1–48, where nucleotide 1 is the first nucleotide of the start codon. For all samples selected after translation of the $NNS_{15}$ library, the region of interest is defined in Figs 1 and S7 and covered nucleotides 24–47. For all ErmBL-related samples, the region of interest covered nucleotides 30–32 from the start codon. A summary of NGS read processing is given in Table S7.

### Analysis of RNase R cleavage in known arrest sequences

Assembled and trimmed reads for all of the known arrest sequences shown in Fig 1B and C were processed using the *process-reads.py* script, with the region of interest covering nucleotides 24–77 and the minimum Q score for base calls within this region set

to 60. We then obtained the size distribution of reads that were exact matches to the 5′ end of each of the sequences in Table S8. This task was automated with our *find-exact-match.py* script.

### Translation into amino acid sequences

Processed reads were translated using the *translate-reads.py* script. For samples that had undergone inverse toeprinting, the ribosome-protected region downstream of the A-site codon was removed before translation. Unique peptidic sequences were identified and the frequency of occurrence of these sequences within each sample was calculated.

### Calculation of 3-aa motif frequencies

All possible 3-aa motifs centered on the P-site of the stalled ribosomes were identified and counted within the translated sequences, using the *process-kmers.py* script with a word size of three. In each case, the frequency of occurrence ($F_{3aa}$) of the 3-aa motif was as follows:

$$F_{3aa} = \frac{n_{3aa}}{n_{total}},$$

where $n_{3aa}$ is the number of occurrences of this 3-aa motif at the C-terminus of the translated sequences and $n_{total}$ is the total number of processed reads in the sample.

### Conversion into codon-based sequences

Processed reads were converted to a codon-based sequence where each of the 64 possible codons was assigned a unique identifier, using the *convert-to-codons.py* script. For samples that had undergone inverse toeprinting, the ribosome-protected region downstream of the A-site codon was removed before translation. Unique codon-based sequences were identified and the frequency of occurrence of these sequences within each sample was calculated.

### Calculation of 3-codon motif frequencies

All possible 3-codon motifs centered on the P-site codon of stalled ribosomes were identified and counted within the codon-based sequences, using the *process-codon-kmers.py* script with a word size of 3 codons. In each case, the frequency of occurrence ($F_{3codon}$) of the 3-codon motif was as follows:

$$F_{3codon} = \frac{n_{3codon}}{n_{total}},$$

where $n_{3codon}$ is the number of occurrences of this 3-codon motif at the C-terminus of the translated sequences, and $n_{total}$ is the total number of processed reads in the sample.

### Calculation of fold changes and propagation of inter-replicate errors

The fold change in 3-aa or 3-codon motif frequency between two samples was calculated using the *read-analyzer.py* script and was defined as $F_{Fg}/F_{Bg}$, where $F_{Fg}$ is the frequency of occurrence of a sequence, 3-aa or 3-aa motif in the "foreground" sample, and $F_{Bg}$ is its frequency in the "background" sample. For the comparison between Ery-treated and untreated samples, the mean frequency

of occurrence of each 3-aa motif in the presence or absence of the drug was as follows:

$$F_{Ery} = \left(F_{Ery1} + F_{Ery2}\right)/2,$$

and

$$F_{NoAb} = \left(F_{NoAb1} + F_{NoAb2}\right)/2,$$

respectively.

Similarly, the errors between replicates were as follows

$$\Delta F_{Ery} = \left| F_{Ery1} - F_{Ery2} \right|,$$

and

$$\Delta F_{NoAb} = \left| F_{NoAb1} - F_{NoAb2} \right|.$$

The fold change in 3-aa motif frequency on addition of Ery was given as $F_{Ery}/F_{NoAb}$, and the combined error of the fold change in 3-aa frequency was as follows:

$$\Delta\left(F_{Ery}/F_{NoAb}\right) = \sqrt{\left(\frac{\Delta F_{NoAb}}{F_{NoAb}}\right)^2 + \left(\frac{\Delta F_{Ery}}{F_{Ery}}\right)^2} \times \frac{F_{Ery}}{F_{NoAb}},$$

The histogram in Fig 3A was built using the $\Delta\left(F_{Ery}/F_{NoAb}\right)$ values for all well-measured 3-aa motifs.

### Calculation of pause strengths

Pause strengths for all 3-aa motifs were calculated using the *calculate_all_pause_strengths.py* script, according to the following formula:

$$\text{Pause strength}_{3aa} = \frac{n_P}{n_P + n_B},$$

where $n_P$ is the number of reads where a ribosome is paused on the 3-aa motif of interest, and $n_B$ is the number of reads where the ribosome has translated through (bypassed) this motif.

# Data Deposition

National Center for Biotechnology Information Short Read Archive: SRP140857.

# Supplementary Information

# Acknowledgments

We thank A Malhotra for providing RNase R, N Vazquez-Laslop, and AS Mankin for providing the pERMZα reporter plasmid, *E. coli* TB1 cells and oleandomycin, and AC Seefeldt for purifying fully modified EF-P. We thank A Buskirk for providing the pause scores for PP-containing motifs calculated from ribosome profiling data. CA Innis, B Seip, and G Sacheau received

funding for this project from the European Research Council under the European Union's Horizon 2020 research and innovation program (grant agreement no. 724040). CA Innis is an EMBO YIP and the recipient of a Marie Curie career integration grant (PCIG14-GA-2013-631479). D Dupuy is funded by Inserm. B Seip, and CA Innis received funding from the Fondation pour la Recherche Médicale (AJE201133), the Région Aquitaine (2012-13-01-009), and a Chaire d'Installation from the excellence initiative (IdEx) of the University of Bordeaux.

## Author Contributions

B Seip: Designed and performed experiments, analyzed data, drafted, reviewed and edited the manuscript.
G Sacheau: Performed experiments, analyzed data, reviewed and edited the manuscript.
D Dupuy: Designed experiments, analyzed data, drafted, reviewed and edited the manuscript.
CA Innis: Conceptualized and supervised the study, designed experiments, analyzed data, drafted, reviewed and edited the manuscript.

## Conflict of Interest Statement

The authors declare that they have no conflict of interest.

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
