## [Reviewer comments · Life Science Alliance]

Life Science Alliance

Ribosomal stalling landscapes revealed by high-throughput inverse toeprinting of mRNA libraries

Britta Seip, Guénaél Sacheau, Denis Dupuy, and Cristobal Innis
DOI: 10.26508/lsa.201800148

Corresponding author(s): Cristobal Innis, European Institute of Chemistry and Biology (IECB) and Denis Dupuy, European Institute of Chemistry and Biology (IECB)

Review Timeline:

Submission Date:	2018-08-07
Editorial Decision:	2018-08-08
Revision Received:	2018-09-25
Accepted:	2018-09-27

Scientific Editor: Andrea Leibfried

Transaction Report:

Please note that the manuscript was previously reviewed at another journal and the reports were taken into account in inviting a revision for publication at *Life Science Alliance* prior to submission to *Life Science Alliance*.

Thank you for transferring your manuscript entitled "Ribosomal stalling landscapes revealed by high-throughput inverse toeprinting of complex transcript libraries" to Life Science Alliance. Your manuscript was previously reviewed at another journal, and the editors have transferred the reports to us.

The reviewers who assessed your work elsewhere noted the high quality of your data, but thought that it remains at this stage unclear whether the described method is applicable in a different context. Based on this input already at hand, we would be happy to publish your paper in Life Science Alliance pending minor revision. We would expect a point-by-point response to the criticisms raised and accordingly text changes. Importantly, the impact and significance of potential applications of the method should be better discussed and the writing adapted to a broader audience. Please let me know in case you have any questions regarding the revision.

REFEREE REPORTS OBTAINED DURING PEER REVIEW ELSEWHERE

Referee #1 Review

Report for Author:

The paper by Seip et al describes a method to identify stalling sequences in transcript libraries. The aim is somewhat narrow but certainly worth exploring. However, the paper is written in a technical language that makes it difficult to assess the broader impact of the method. While the authors provide extensive validation, the principles of experiments are not described well and the applicability of the method for tasks other than to determine stalling events remains unclear. There is little biological novelty in the paper, although I understand that this was not the aim.

Specifically, I have the following questions:

1. It is absolutely unclear how the "complex transcript libraries" are defined and generated. It seems that the design of such libraries requires accurate knowledge of the stalling mechanism and targets. What are the chances to construct unbiased libraries? Which validation experiments were carried out using unbiased libraries? What is the sequence depth that one can achieve?
2. What are the limitations of the PURE translation system used by the authors with respect to number of mRNA sequences that can be translated, in particular in the situation where ribosome turnover is limited by stalling? Aren't these experiments simply prohibitively expensive to make a truly open search for unknown sequences? Does this experimental setup allow to test genome information from other organisms (I guess not).
3. What is the range of questions that can be addressed by the method except for stalling (which is studied quite well by different methods)?

Minor points:

1. In the text, the authors state that they avoid RF2, but in the Fig it is RF1/RF3. Please clarify.
2. Please explain the concept of "pause strength". Which pause is stronger, with pause strength 0.2 or 0.8? For validation, it would be very useful to report pause strengths for established stalling sequences, e.g. SecM.
3. It remains unclear why the authors test the combination of polyproline and antibiotic

stalling - does it make any biological sense?

4. What is the arrest motif XP(C), is there a cysteine in the 3rd position?

5. In Fig 3, the controls for most XPP and PPX sequences + EF-P are missing, this has to be provided.

6. In Fig 3, the result shown in f and g are not properly described and discussed.

Referee #2 Review

Report for Author:

The manuscript entitled "Ribosomal stalling landscapes..." by Seip et al describes a new method that the authors call 'high-throughput inverse toeprinting' and that allows them to identify peptide-encoding transcripts that induce ribosomal stalling in vitro. Unlike ribosome profiling, high-throughput inverse toeprinting preserves and allows sequencing of the mRNA sequence that is upstream of the stalled ribosome. Unlike classical toeprinting, it is amenable to high-throughput, next generation sequencing. Thus, high-throughput inverse toeprinting provides information and advantages that are distinct from, but yet highly complementary to, ribosome profiling and classical toeprinting. To validate and demonstrate the effectiveness of high-throughput inverse toeprinting, Seip et al used this approach to characterize the stalling landscapes of free and antibiotic-bound bacterial ribosomes in the context of specific candidate mRNAs (the ErmBL, ErmAL1, ErmCL, ErmDL, SecM, and TnaC mRNAs) as well as in the context of an mRNA library (the NNS15 library). In the case of the candidate mRNAs, Seip et al were not only able to identify known stalling sites, but they were also able to identify a new stalling site in the ErmAL1 mRNA. In the case of the mRNA library, high-throughput inverse toeprinting experiments performed in the absence of antibiotics and the stalled-ribosome rescue factor, EF-P, enabled Seip et al to quantify the intrinsic 'pause strength' of many three-amino acid motifs. The analysis of these data demonstrates the authors' ability to identify strong, intrinsic stalling sites that have been previously identified using biochemical methods and ribosome profiling. Repeating these experiments using the mRNA library in the presence of erythromycin again identified three-amino acid motifs that have been previously identified as erythromycin-dependent stalling sites, but also revealed at least one new stalling site; the authors also identified which of these erythromycin-dependent stalling events could be rescued or, in new findings, exacerbated by EF-P. Finally, the authors used high-throughput inverse toeprinting of a library of ErmBL mRNA variants in the absence of antibiotics, the presence of erythromycin, or the presence of the related antibiotic, oleandomycin, to identify ErmBL variants that would exhibit differential stalling behavior in the presence of erythromycin versus oleandomycin. Together with an in vivo stalling assay, these experiments enabled Seip et al to identify ErmBL variants that exhibit stalling in the presence of erythromycin, but not in the presence of oleandomycin.

The design of the experiments in this study is appropriate, the data are of a high quality and have been carefully analyzed, and the conclusions drawn by the authors are well-supported by the data. Overall, the high-throughput inverse toeprinting method described, validated, and applied by the authors is a new, high-throughput tool that merges the advantages of ribosome profiling and classical toeprinting, but that is able to provide information that cannot be obtained using ribosome profiling or that cannot be obtained in a large-scale manner using classical toeprinting. Although the major impact of high-throughput inverse toeprinting seems to be limited to a relatively small set of specialized situations (e.g., when a reference genome is not available, when using random or focused mRNA sequence libraries, and/or when conducting systematic screens for identifying drug-dependent stalling sites), I expect that it will become an important tool for these specialized uses. As a proof-of-principle, the authors demonstrate their ability to identify ErmBL variant sequences that exhibit stalling in the

presence of erythromycin but not oleandomycin, a finding that would have been extremely difficult to make using ribosome profiling or classical toeprinting. Thus, assuming that the authors can address the minor comments listed below, I would recommend publication of this manuscript in the journal.

Minor Comments

1. The authors do a very good job of highlighting the differences between their high-throughput inverse toeprinting method and the ribosome profiling and classical toeprinting methods. They also do a good job of identifying the specialized situations in which high-throughput inverse toeprinting will be able to provide unique, complementary, or comparable information relative to what can be obtained using ribosome profiling or classical toeprinting or information. What is lacking, in my view, is a description of 1-2 specific, definitive, high-impact examples of specialized situations in which high-throughput inverse toeprinting would be able to provide unique, actionable information that cannot be obtained by other methods. As an example of what I mean, the proof-of-principle identification of ErmBL variant sequences that exhibit stalling in the presence of erythromycin but not oleandomycin seems to be a solid result that could only have been easily obtained using high-throughput inverse toeprinting. However, the authors do not clearly state what the impact of this finding is. How significant is it to be able to identify an erythromycin-dependent stalling site that is resistant to oleandomycin-dependent stalling? More generally, how significant is it to be able to identify a stalling site that is specifically sensitive to one antibiotic versus a closely related antibiotic. I'm not saying that it isn't significant, but only that the authors have not articulated the significance. This is true of the other examples that the authors' list when they describe the advantages of high-throughput inverse toeprinting. In general, they should articulate the impact and significance of potential applications more clearly.
2. On page 3, in the first subsection of the results, the authors should explicitly state whether the experiments performed using the ErmBL mRNA were performed in the absence and/or presence of erythromycin. Likewise, they should explicitly state whether the follow-up experiments performed using the ErmAL1, ErmCL, ErmDL, SecM, and TnaC mRNAs were performed in the absence and/or presence of the corresponding antibiotic or ligand.
3. On page 4, the authors state that they size-selected inverse toeprints in order to minimize contamination from inverse toeprints arising from initiation complexes. A close look at Supplementary Fig 3 demonstrates that the majority of the inverse toeprints arise from initiation complexes and are therefore excluded from the sequencing and analysis. Why are so many of the mRNAs apparently stalled in initiation complexes? Does their exclusion bias the sequencing, analyses, findings, and/or interpretations? The authors should discuss this in the manuscript.
4. On page 5, the authors state that they can precisely and reproducibly measure the frequency of about two-thirds of the 8,000 possible 3-aa motifs in their high-throughput inverse toeprinting experiments using the mRNA library. I presume that the remaining one-third are underrepresented in the library. Is that correct? If so, why is that? Regardless, does the fact that one-third of the possible 3-aa motifs are missing from the results and analysis limit or bias the analysis, findings, and/or interpretations in any way? The authors should discuss this in the manuscript as well.

Referee #3 Review

Report for Author:

Overall assessment:

The manuscript by Britta Seip et al. entitled 'Ribosomal stalling landscapes revealed by high-throughput inverse toeprinting of complex transcript libraries' reports on the development and application of an elegant strategy called inverse toeprinting enabling the in vitro delineation of the mRNA region upstream of a stalled ribosome with codon resolution. In their study, inverse toeprinting was used to examine (changes in) stalling landscapes of free and drug-bound *Escherichia coli* ribosomes, enabling the investigation of ribosomal stalling by nascent peptides by making use of random and focused transcript libraries.

Overall assessment:

The manuscript is written in a clear way and the research context sufficiently documented. While the incremental benefit of the use of inverse toeprinting to study ribosome stalling still needs to be proven when compared to the use of alternative in vivo approaches such as ribosome profiling, the data analyses performed convincingly hints to the implication of a comprehensive list of arrest motifs, of which the strengths of translational pausing were found to correlate with in vivo stalling sequences. As such, the authors nicely demonstrated the validity of their approach.

Comments:

- Besides the enrichment of 3-AA motifs, it would be informative to look at the enrichment of codons/nucleotide sequences, to determine if the arrest observed is only dependent of AA motifs and if this is influenced by the redundancy of codon usage.
- Since the study only focuses on AA motif enrichment of nascent chains, the (putative) involvement of other causative factors of ribosome stalling (e.g. secondary mRNA structures) should also be discussed. Further, do the authors believe that the latter are causative for the (modest) discrepancies observed between in vitro and in vivo profiling data?
- Pg. 10 - discussion; specify more clearly what is meant with the discrepancies due to the intrinsic focus on the early cycles of translation.
- Clarify what is meant with NNS library upon first mentioning.

Referee #1:

The paper by Seip et al describes a method to identify stalling sequences in transcript libraries. The aim is somewhat narrow but certainly worth exploring. However, the paper is written in a technical language that makes it difficult to assess the broader impact of the method. While the authors provide extensive validation, the principles of experiments are not described well and the applicability of the method for tasks other than to determine stalling events remains unclear. There is little biological novelty in the paper, although I understand that this was not the aim.

We thank the referee for their review of our work and have made changes to our manuscript (especially the discussion section) in order to make the impact of our work clearer.

Specifically, I have the following questions:

1. It is absolutely unclear how the "complex transcript libraries" are defined and generated.

Library complexity refers to the number of unique molecules (or sequence variants) that are sampled by sequencing. In our work, the phrase "complex transcript libraries" was intended to refer to libraries composed of millions of unique sequence variants, of which 10^5 - 10^6 would be sampled by next-generation sequencing. We agree that the use of this phrase does not precisely describe how the libraries are defined. As a result, we have removed it from the title and replaced it with "transcript libraries of any given complexity" on p.4 or with "high complexity library" on p. 8.

How the libraries were generated is explained in section 1.2 ("Experimental Procedures") of the Methods section. In addition, WebLogos for the NNS₁₅ and ErmBL libraries are shown in supplementary Figure 2. As the nucleotide frequency at each position of the *NNS15* and *ermBL_deep_mutated* oligonucleotides was not provided in the original version of our manuscript, we have included this information in Supplementary Table 3.

It seems that the design of such libraries requires accurate knowledge of the stalling mechanism and targets. What are the chances to construct unbiased libraries? Which validation experiments were carried out using unbiased libraries?

The design of the libraries used with inverse toeprinting need not require prior knowledge of the stalling mechanism and can be made in a totally unbiased manner. The NNS₁₅ library encodes random 20-residue peptides, each comprising 15 consecutive amino acids coded for by NNS codons, where N refers to any nucleotide (in equal proportions) and S refers to C or G (also in equal proportions). We chose to use NNS codons to avoid the presence of UAA and UGA stop codons that would be recognized by RF-2, which is omitted from the translation reaction. All of the validation experiments described on p. 5-7 were performed using this unbiased library.

What is the sequence depth that one can achieve?

The depth of sequencing that can be achieved will depend on the scale of the next-generation sequencing run chosen. For this study, the sequencing depth is given in supplementary Table 7. The depth required to achieve reproducible results is more than one order of magnitude lower than for ribosome profiling, meaning that more conditions can be tested for the same cost.

2. What are the limitations of the PURE translation system used by the authors with respect to number of mRNA sequences that can be translated, in particular in the situation where ribosome turnover is limited by stalling? Aren't these experiments simply prohibitively expensive to make a truly open search for unknown sequences? Does this experimental setup allow to test genome information from other organisms (I guess not).

10^{11} - 10^{12} mRNAs can be tested in a 10 μ L in vitro translation reaction using the PURE system. Since writing this manuscript, we have shown in the lab that 5 μ L of reaction are sufficient. We are now in the process of using inverse toeprinting for hundreds of parallel reactions in order to identify novel metabolite-dependent arrest peptides. This is carried out at a fraction of a cost of what it would take to perform hundreds of ribosome profiling reactions. Testing genome information from other organisms is possible by replacing the *E. coli* ribosomes in the PURExpress system with ribosomes from other species. This may not work for all species, but has been successfully achieved with *Bacillus* or *Thermus* ribosomes for example.

3. What is the range of questions that can be addressed by the method except for stalling (which is studied quite well by different methods)?

We think that inverse toeprinting could be adapted to the study of any sequence-dependent biological process that can be linked to ribosomal pausing on the mRNA. This includes co-translational membrane targeting and insertion, transcriptional or translational regulation by proteins, RNAs or small molecules that act as roadblocks on the mRNA, or factors that modulate the activity of the ribosome in a sequence-dependent manner. Since this would require additional modifications to our method, we have decided that listing these applications is beyond the scope of this study and have modified the discussion section accordingly.

Inverse toeprinting does, however, provide a solution for a number of problems related to ribosome stalling that are not suitably addressed by existing methods, as we have explained in the introduction and discussion sections of the manuscript.

Minor points:

1. In the text, the authors state that they avoid RF2, but in the Fig it is RF1/RF3. Please clarify.

We thank the reviewer for pointing out this mistake. Fig. 1 has been modified accordingly.

2. Please explain the concept of "pause strength". Which pause is stronger, with pause strength 0.2 or 0.8? For validation, it would be very useful to report pause strengths for established stalling sequences, e.g. SecM.

We added a few sentences on p. 6 to better explain the concept of pause strength. The pause strength of established stalling sequences like SecM cannot be calculated because the formula for obtaining it requires both the number of times ribosomes are seen on the motif and the number of times the motif is bypassed. For a motif longer than 3-4 amino acids, there are simply not enough occurrences of the bypassed motif and all pause strengths calculated would therefore be 1. However, we show in Figure 2c that there is a strong correlation between the pause strength of a motif and the extent to which it is enriched following selection. Thus, the $\log_2(\text{fold change})$ in frequency of a motif can be used as a proxy for pause strength.

3. It remains unclear why the authors test the combination of polyproline and antibiotic stalling - does it make any biological sense?

The interaction between polyproline-mediated and antibiotic-dependent stalling is an unexpected result that was detected by our unbiased approach and not something that we set out to test specifically. The reason why XP(X) motifs stall the ribosome more efficiently in the presence of EF-P than in its absence is unclear at the moment, as is the increased stalling efficiency of PP(X) motifs in the presence of Ery. If investigated further, this phenomenon may yield some additional insights into the mechanism of polyproline stalling and rescue, but we do not currently have biochemical data to build upon this initial observation. As a result, we decided to remove all references to the samples containing EF-P in Fig 3 (panels F and G now form supplementary Fig S1 I) and shortened the discussion of these results on p. 7.

4. What is the arrest motif XP(C), is there a cysteine in the 3rd position?

The residue in parentheses indicates the incoming amino acid in the A-site. Hence ribosomes pausing on XP(C) motifs have a Pro codon in the P-site and a Cys codon in the A-site. This is now explained on p. 6.

5. In Fig 3, the controls for most XPP and PPX sequences + EF-P are missing, this has to be provided.

The discussion of the effect of EF-P on arrest at XP(X) motifs was relegated to supplementary material to streamline the manuscript. As a result, we removed all reference to the samples containing EF-P in Fig 3. All data obtained in the presence of EF-P (including the controls requested) are now in supplementary Fig S1 I.

6. In Fig 3, the result shown in f and g are not properly described and discussed.

We have now moved these panels to the supplementary materials as we think these results are worth reporting, but we do not have sufficient additional data to make sense of them. Consequently, the discussion of these results has been kept to a minimum.

Referee #2:

The manuscript entitled "Ribosomal stalling landscapes..." by Seip et al describes a new method that the authors call 'high-throughput inverse toeprinting' and that allows them to identify peptide-encoding transcripts that induce ribosomal stalling in vitro. Unlike ribosome profiling, high-throughput inverse toeprinting preserves and allows sequencing of the mRNA sequence that is upstream of the stalled ribosome. Unlike classical toeprinting, it is amenable to high-throughput, next generation sequencing. Thus, high-throughput inverse toeprinting provides information and advantages that are distinct from, but yet highly complementary to, ribosome profiling and classical toeprinting. To validate and demonstrate the effectiveness of high-throughput inverse toeprinting, Seip et al used this approach to characterize the stalling landscapes of free and antibiotic-bound bacterial ribosomes in the context of specific candidate mRNAs (the ErmBL, ErmALI, ErmCL, ErmDL, SecM, and TnaC mRNAs) as well as in the context of an mRNA library (the NNS15 library). In the case of the candidate mRNAs, Seip et al were not only able to identify known stalling sites, but they were also able to identify a new stalling site in the ErmALI mRNA. In the case of the mRNA library, high-throughput inverse toeprinting experiments performed in the absence of antibiotics and the stalled-ribosome rescue factor, EF-P, enabled Seip et al to quantify the intrinsic 'pause strength' of many three-amino acid motifs. The analysis of these data demonstrates the authors' ability to identify strong, intrinsic stalling sites that have been previously identified using biochemical methods and

ribosome profiling. Repeating these experiments using the mRNA library in the presence of erythromycin again identified three-amino acid motifs that have been previously identified as erythromycin-dependent stalling sites, but also revealed at least one new stalling site; the authors also identified which of these erythromycin-dependent stalling events could be rescued or, in new findings, exacerbated by EF-P. Finally, the authors used high-throughput inverse toeprinting of a library of ErmBL mRNA variants in the absence of antibiotics, the presence of erythromycin, or the presence of the related antibiotic, oleandomycin, to identify ErmBL variants that would exhibit differential stalling behavior in the presence of erythromycin versus oleandomycin. Together with an *in vivo* stalling assay, these experiments enabled Seip et al to identify ErmBL variants that exhibit stalling in the presence of erythromycin, but not in the presence of oleandomycin.

The design of the experiments in this study is appropriate, the data are of a high quality and have been carefully analyzed, and the conclusions drawn by the authors are well-supported by the data. Overall, the high-throughput inverse toeprinting method described, validated, and applied by the authors is a new, high-throughput tool that merges the advantages of ribosome profiling and classical toeprinting, but that is able to provide information that cannot be obtained using ribosome profiling or that cannot be obtained in a large-scale manner using classical toeprinting. Although the major impact of high-throughput inverse toeprinting seems to be limited to a relatively small set of specialized situations (e.g., when a reference genome is not available, when using random or focused mRNA sequence libraries, and/or when conducting systematic screens for identifying drug-dependent stalling sites), I expect that it will become an important tool for these specialized uses. As a proof-of-principle, the authors demonstrate their ability to identify ErmBL variant sequences that exhibit stalling in the presence of erythromycin but not oleandomycin, a finding that would have been extremely difficult to make using ribosome profiling or classical toeprinting. Thus, assuming that the authors can address the minor comments listed below, I would recommend publication of this manuscript in the journal.

Minor Comments

1. The authors do a very good job of highlighting the differences between their high-throughput inverse toeprinting method and the ribosome profiling and classical toeprinting methods. They also do a good job of identifying the specialized situations in which high-throughput inverse toeprinting will be able to provide unique, complementary, or comparable information relative to what can be obtained using ribosome profiling or classical toeprinting or information. What is lacking, in my view, is a description of 1-2 specific, definitive, high-impact examples of specialized situations in which high-throughput inverse toeprinting would be able to provide unique, actionable information that cannot be obtained by other methods. As an example of what I mean, the proof-of-principle identification of ErmBL variant sequences that exhibit stalling in the presence of erythromycin but not oleandomycin seems to be a solid result that could only have been easily obtained using high-throughput inverse toeprinting. However, the authors do not clearly state what the impact of this finding is. How significant is it to be able to identify an erythromycin-dependent stalling site that is resistant to oleandomycin-dependent stalling? More generally, how significant is it to be able to identify a stalling site that is specifically sensitive to one antibiotic versus a closely related antibiotic. I'm not saying that it isn't significant, but only that the authors have not articulated the significance. This is true of the other examples that the authors' list when they describe the advantages of high-throughput inverse toeprinting. In general, they should articulate the impact and significance of potential applications more clearly.

We thank the reviewer for their careful review of our work and for their insightful comments.

A section describing some of the questions that could be addressed by the method has now been added to the discussion section. We hope that this will better convey the significance and potential impact of our method.

2. On page 3, in the first subsection of the results, the authors should explicitly state whether the experiments performed using the ErmBL mRNA were performed in the absence and/or presence of erythromycin. Likewise, they should explicitly state whether the follow-up experiments performed using the ErmALI, ErmCL, ErmDL, SecM, and TnaC mRNAs were performed in the absence and/or presence of the corresponding antibiotic or ligand.

We have modified the text on p. 3 and in the legend for Figure 1 to explicitly state the above.

3. On page 4, the authors state that they size-selected inverse toeprints in order to minimize contamination from inverse toeprints arising from initiation complexes. A close look at Supplementary Fig 3 demonstrates that the majority of the inverse toeprints arise from initiation complexes and are therefore excluded from the sequencing and analysis. Why are so many of the mRNAs apparently stalled in initiation complexes? Does their exclusion bias the sequencing, analyses, findings, and/or interpretations? The authors should discuss this in the manuscript.

We have added some text to p. 6 to briefly discuss this. The bottom line is that we always observe inverse toeprints corresponding to initiation complexes, but their abundance varies depending on the lot of PURExpress system used. We think that this corresponds to the peaks of increase ribosome density seen at the beginning of coding regions by ribosome profiling, but that it is further exaggerated by inefficiencies of the PURExpress system. All of the experiments in this study were performed using the same lot of PURExpress system and removal of these inverse toeprints allowed us to obtain more reads for the longer inverse toeprints without biasing our analysis.

4. On page 5, the authors state that they can precisely and reproducibly measure the frequency of about two-thirds of the 8,000 possible 3-aa motifs in their high-throughput inverse toeprinting experiments using the mRNA library. I presume that the remaining one-third are underrepresented in the library. Is that correct? If so, why is that? Regardless, does the fact that one-third of the possible 3-aa motifs are missing from the results and analysis limit or bias the analysis, findings, and/or interpretations in any way? The authors should discuss this in the manuscript as well.

Poorly measured motifs are sequenced fewer than 150 times when the two replicates after selection are combined. Since the vast majority of motifs are well represented in the input library (i.e. only 11 motifs have < 150 reads in the sequenced NNS15 library), poorly measured motifs correspond to motifs that become depleted during the selection process because they do not induce pauses in translation. Consequently, their removal does not limit or bias the analysis. This is now discussed on p. 6 of the manuscript.

Referee #3:

Overall assessment:

The manuscript by Britta Seip et al. entitled 'Ribosomal stalling landscapes revealed by high-throughput inverse toeprinting of complex transcript libraries' reports on the development and application of an elegant strategy called inverse toeprinting enabling the in vitro delineation of the mRNA region upstream of a stalled ribosome with codon resolution. In their study, inverse toeprinting was used to examine (changes in) stalling landscapes of free and drug-bound

Escherichia coli ribosomes, enabling the investigation of ribosomal stalling by nascent peptides by making use of random and focused transcript libraries.

Overall assessment:

The manuscript is written in a clear way and the research context sufficiently documented.

While the incremental benefit of the use of inverse toeprinting to study ribosome stalling still needs to be proven when compared to the use of alternative *in vivo* approaches such as ribosome profiling, the data analyses performed convincingly hints to the implication of a comprehensive list of arrest motifs, of which the strengths of translational pausing were found to correlate with *in vivo* stalling sequences. As such, the authors nicely demonstrated the validity of their approach.

We thank the reviewer for their careful review and for appreciating the validity of our work.

Comments:

- Besides the enrichment of 3-AA motifs, it would be informative to look at the enrichment of codons/nucleotide sequences, to determine if the arrest observed is only dependent of AA motifs and if this is influenced by the redundancy of codon usage.

We looked at the enrichment of 3-codon motifs and found no impact of the nucleotide sequence on the arrest process under our *in vitro* conditions (see the new supplementary Fig S6 and additional text on p.7). Although we cannot generalize this to what occurs *in vivo*, these results show the usefulness of our method in measuring the effect of peptide-induced pauses in translation rather than detecting inefficient translation caused by consecutive rare codons.

- Since the study only focuses on AA motif enrichment of nascent chains, the (putative) involvement of other causative factors of ribosome stalling (e.g. secondary mRNA structures) should also be discussed. Further, do the authors believe that the latter are causative for the (modest) discrepancies observed between *in vitro* and *in vivo* profiling data?

Information concerning the possible impact of mRNA secondary structures downstream of the stalled ribosome is lost due to RNase R digestion. As a result, we cannot comment on the impact these secondary structures have on the pausing and whether they are responsible for the small discrepancies observed between the *in vitro* and *in vivo* data.

- Pg. 10 - discussion; specify more clearly what is meant with the discrepancies due to the intrinsic focus on the early cycles of translation.

We have added a sentence to this effect on p. 9.

- Clarify what is meant with NNS library upon first mentioning.
We have added a sentence on p. 6 to explain this.

Thank you for submitting your revised manuscript entitled "Ribosomal stalling landscapes revealed by high-throughput inverse toeprinting of mRNA libraries". I appreciate the introduced changes, and it is a pleasure to let you know that your manuscript is now accepted for publication in Life Science Alliance. Congratulations on this interesting work.